# The Separation and Purification of Ellagic Acid from *Phyllanthus urinaria* L. by a Combined Mechanochemical-Macroporous Resin Adsorption Method

**Zili Guo** , **Shuting Xiong, Yuanyuan Xie and Xianrui Liang** *

Key Laboratory for Green Pharmaceutical Technologies and Related Equipment of the Ministry of Education, College of Pharmaceutical Sciences, Zhejiang University of Technology, Hangzhou 310014, China; guozili@zjut.edu.cn (Z.G.); shutingx12@163.com (S.X.); xyycz@zjut.edu.cn (Y.X.)
* Correspondence: liangxrvicky@zjut.edu.cn

**Abstract:** Ellagic acid is a phenolic compound that exhibits both antimutagenic and anticarcinogenic activity in a wide range of assays in vitro and in vivo. It occurs naturally in some foods such as raspberries, strawberries, grapes, and black currants. In this study, a valid and reliable method based on mechanochemical-assisted extraction (MCAE) and macroporous adsorption resin was developed to extract and prepare ellagic acid from *Phyllanthus urinaria* L. (PUL). The MCAE parameters, acidolysis, and macroporous adsorption resin conditions were investigated. The key MCAE parameters were optimized as follows: the milling time was 5 min, the ball mill speed was 100 rpm, and the ball mill filling rate was 20.9%. Sulfuric acid with a concentration of 0.552 mol/L was applied for the acidolysis with the optimized acidolysis time of 30 min and acidolysis temperature of 40 °C. Additionally, the XDA-8D macroporous resin was chosen for the purification work. Both the static and dynamic adsorption tests were carried out. Under the optimized conditions, the yield of ellagic acid was 10.2 mg/g, and the content was over 97%. This research provided a rapid and efficient method for the preparation of ellagic acid from the cheaply and easily obtained PUL. Meanwhile, it is relatively low-cost work that can provide a technical basis for the comprehensive utilization of PUL.

**Keywords:** *Phyllanthus urinaria* L.; ellagic acid; mechanochemical-assisted extraction; macroporous adsorption resin

## 1. Introduction

Ellagic acid ($C_{14}H_6O_8$, Figure 1) is an acidic hydrolysis product of polymeric ellagitannins. It is a natural polyphenolic compound that is present in various vegetables, fruits, herbs, and nuts, such as raspberries, strawberries, walnuts, grapes, and black currants [1,2]. At present, the most important pharmacological activity of ellagic acid is its antioxidant activity, which mainly depends on its basic structure containing four hydroxyl groups that are responsible for scavenging superoxide and hydroxyl anion free radicals [3–5]. In addition, ellagic acid has been found to have antimutagenic [6], anticarcinogenic [7–11], antifibrosis [9], and anti-inflammatory effects [12,13]. Furthermore, it can also improve Alzheimer's disease-mediated dementia by suppressing oxidative and inflammatory cell damage and improving antioxidant content. Ellagic acid inhibits cognitive abnormality following traumatic brain injury (TBI) in rats through its anti-inflammatory and antioxidant properties [14]. At present, three main methods are utilized for preparing ellagic acid: plant extraction [15,16], chemical synthesis [17], and enzymatic degradation [18]. Among them, the extraction of ellagic acid from natural plant resources is thought to be one of the most important ways due to its relatively high abundant content and environmental friendliness.

**Figure 1.** The chemical structure of ellagic acid.

*Phyllanthus urinaria* L. (PUL), belonging to the *Euphorbiaceae* family, is widely distributed in tropical and subtropical regions of Asian countries [19,20]. The dried whole plant of PUL, also named "pearl grass", "nocturnal grass", and "yin-yang grass", is a traditional Chinese medicine for the treatment of several diseases, including dysentery, jaundice, urinary tract infection, and malnutritional stagnation [21–23]. Pharmacological studies have shown that PUL possesses good biological activities such as anti-viral [24–26], anti-tumor [27], anti-thrombosis [28], hepatoprotective [29], antioxidant [30,31], anti-inflammatory [32] and immunomodulatory activities [33]. Most of the reported ingredients in PUL were tannins, lignans, flavonoids, phenolics, and terpenoids [23,29,30]. Ingredient research also showed that PUL is rich in ellagitannins, which are the main anti-hepatitis B virus active ingredients [22,24] and the main sources of ellagic acid.

In this work, a reliable method based on mechanochemical-assisted extraction (MCAE) and macroporous adsorption resin was developed to extract, separate and purify ellagic acid from PUL. To promote the yield of ellagic acid, the key mechanochemistry ball milling parameters were optimized. Additionally, the different kinds of macroporous resin were further optimized to purify the ellagic acid by the static and dynamic adsorption tests. To date, no separation of ellagic acid from PUL based on mechanochemical-assisted extraction and macroporous adsorption resin has been reported. This research provided both an efficient method to obtain ellagic acid and the comprehensive utilization of cheaply and easily obtained PUL.

## 2. Materials and Methods

### 2.1. Materials and Chemicals

The dried PUL was purchased from Tongrentang, Hangzhou City, Zhejiang Province (Hanghzou, China) and stored at room temperature before analysis. The standard ellagic acid (≥98%) was purchased from Shanghai Yuanye Bio-Technology Co., Ltd. (Shanghai, China).

HPLC-grade acetonitrile, methanol and ethanol were supplied by Merck (Darmstadt, Germany). HPLC-grade formic acid was bought from Shanghai Aladdin Bio-Chem Technology Co., Ltd. (Shanghai, China). Ultrapure water (18.2 MΩ) was purified by Barnstead TII super Pure Water System (Boston, MA, USA). Other reagents used in this experiment were all analytical grade and were obtained from Yongda Chemical Reagent Company (Tianjin, China).

### 2.2. Mechanochemical-Assisted Extraction Procedure

The dried PUL was fully ground and passed through an 80-mesh sifter. The PUL powder (2.0 g) was added into a PM 200 planetary ball mill (grinding media: stainless steel balls of 8 mm diameter; the weight of the balls: 4.2 g; two drums at 50 mL each; the volume of the load/drum ratio: 1:2). Then, the ground mixture (1.0 g) was extracted with 10 mL of 50% ethanol in a 50 mL flask at 30 °C for 30 min under an ultrasonic bath. The supernatant was collected as the ellagitannin extract.

### 2.3. Acidolysis Experiment

The acid hydrolysis agent (1.5 mL) was added to the ellagitannin extract. The mixture was placed in a 40 °C water bath, heated and stirred for 30 min. The supernatant was

concentrated by decompression evaporation and freeze-dried. After removing the solvent, crude ellagic acid was obtained.

### 2.4. Macroporous Resin Adsorption Experiment

### 2.4.1. Pretreatment of Adsorbents

Ten macroporous resins named XAD-2, HP-20, AB-8, XDA-8D, LSA-8D, HPD450, HPD826, DA201, LXA-8, and LX-8 were investigated. Their physical properties were listed in Table 1. To remove some water-soluble impurities and suspended material, resins were soaked in deionized water. Then, they were transferred into 95% ethanol for 24 h to swell completely. After washing with deionized water, resins were separately immersed in 5% (*w/v*) NaOH for 24 h, followed by rinsing with deionized water to a neutral pH. Finally, the macroporous resins were immersed in 5% (*v/v*) HCl for 24 h and rinsed with deionized water again.

**Table 1.** Physical properties of the tested resins in this study.

| Trade Name | Specific Surface Area (m²/g) | Particle Size (mm) | Polarity Type |
|---|---|---|---|
| XAD-2 | 300 | 0.25–0.84 | Non-polarity |
| HP-20 | 590 | 0.25–0.60 | Non-polarity |
| AB-8 | 480–520 | 0.30–1.25 | Weak polarity |
| XDA-8D | 140 | 0.20–0.40 | Medium polarity |
| LSA-8D | 150 | 0.30–1.25 | Medium polarity |
| HPD450 | 500–550 | 0.30–1.25 | Medium polarity |
| HPD826 | 500–600 | 0.30–1.25 | Medium polarity |
| DA201 | 150–200 | 0.30–1.25 | Polarity |
| LXA-8 | 200 | 0.30–1.25 | Polarity |
| LX-8 | 1000 | 0.315–1.26 | Polarity |

### 2.4.2. Static Adsorption Tests

Different kinds of resins were compared for their separation capacity through static adsorption tests. The general experimental procedure was as follows: Pretreated resin (1.0 g) was added to a 50 mL flask, then 20.0 mL of 0.09 mg/mL crude ellagic acid solution was added. The flask was shaken in a shaker at 25 °C with 100 rpm for 24 h. The content of the ellagic acid in the adsorption solution was determined and calculated by the UPLC method.

### 2.4.3. Sorption Kinetics Tests

Sorption kinetics tests were performed to choose the most efficient resin. Similarly, pretreated resin (1.0 g) was added to a 50 mL flask, then 20.0 mL of 0.09 mg/mL crude ellagic acid solution was added. The flask was shaken in a shaker at 25 °C with 100 rpm for 24 h. Every hour, 1 mL of extract was pipetted out to perform UPLC determination.

The adsorption properties including the adsorption capacity, adsorption rate, desorption ratio and recovery rate of each resin were quantified according to the following equations:

$$W \text{ (mg/g dry resin)} = (\rho_o - \rho_e) \times V/m, \tag{1}$$

$$E \, (\%) = (\rho_o - \rho_e)/\rho_o \times 100\%, \tag{2}$$

$$D \, (\%) = \rho_d/(\rho_o - \rho_e) \times 100\%, \tag{3}$$

$$R \, (\%) = (\rho_o - \rho_e) \times V/m, \tag{4}$$

where W was the adsorption capacity at adsorption equilibrium (mg/g dry resin), and $\rho_o$, $\rho_e$ and $\rho_d$ were the initial, absorption equilibrium and desorption concentrations of analyte in the solutions, respectively (mg/mL). V was the volume of the adsorption solution (mL),

and m was the dry weight of the resin (g). E was the adsorption rate (%), D was the desorption rate (%) and R was the recovery rate (%) of the resin.

### 2.4.4. Dynamic Adsorption Tests

Dynamic adsorption tests were carried out as follows: 60 mL of XDA-8D resins were packed in the column (20 mm × 300 mm), which was loaded with the crude ellagic acid solution, by the wet method. Then, 120 mL of deionized water was loaded to rinse the resins with the flow rate at 1.0 BV/h to make the eluant and extract mix together. After adsorptive equilibrium, different concentrations of ethanol (10%, 20%, 30%, 40%, 50%, 60%, 70%, 80% and 90%) were utilized to desorb the ellagic acid at a constant flow rate of 1.0 BV/h. The elution volume of each concentration was constant by being maintained at 3.0 BV. The contents of ellagic acid in each desorption solution were determined by UPLC.

### 2.5. Ultra High-Performance Liquid Chromatography (UPLC): Quantitative Analysis and the Characterization of Ellagic Acid

The quantitative analysis of ellagic acid was calculated based on the standard curve of ellagic acid by an ACQUITY UPLC$^{TM}$ system (Waters, Milford, MA, USA) equipped with a binary solvent pump, an autosampler, an integral column heater and a photodiode array detector (PDA eλ Detector). The separation was operated on a Waters ACQUITY UPLC HSS T3 column (2.1 mm × 100 mm, 1.8 μm, Waters, Milford, MA, USA). The mobile phase consisted of 0.1% (*v/v*) formic acid solution (A) and acetonitrile (B) with the following gradient: 0~3.5 min, 3–4% B; 3.5~5.0 min, 4–8% B; 5.0~9.0 min, 8–10% B; 9.0~14.0 min, 10–11% B; 14.0~38.0 min, 11–15% B; 38.0~46.0 min, 15–20% B; 46.0~50.0 min, 20–30% B; 50.0~55.0 min, 30–40% B; 55.0~57.0 min, 40–90% B; 57.0~60.0 min, 90% B. The flow rate of the mobile phase was set at 0.2 mL/min. The injection volume was 1.0 μL, and the column temperature was maintained at 30 °C. The wavelength at 250 nm was set as the monitoring wavelength.

The proton $^1$H and carbon $^{13}$C NMR spectra of the ellagic acid were obtained at 308 K using a Bruker AVANCE III 600 MHz NMR spectrometer (Bruker, Billerica, MA, USA). The chemical shifts (δ) in the NMR spectra were recorded in ppm with the solvent peak as the reference. The MS analysis was carried out on a micrOTOF-Q II mass spectrometer (Bruker Daltonics, Bremen, Germany) equipped with an ESI source. The ESI source parameters were as follows: the dry gas ($N_2$) flow rate was 6.0 L/min, the nebulizer gas ($N_2$) pressure was 0.8 bar, the dry gas temperature was 200 °C, and the capillary voltage was 2800 V in the negative mode.

## 3. Results and Discussion

### 3.1. Optimization of the MCAE Procedure

The mechanical force could reduce the particle size of herbal plant powder and destroy the cell wall structure, which increases the release of chemical constituents. In this research, the mechanochemistry ball milling method was applied to the extraction of ellagitannin from PUL. Compared with the traditional extraction method, the ball milling process could effectively increase the peak area of ellagitannin compounds.

To obtain a higher yield of ellagitannin, the key ball milling parameters including milling time, milling speed, and mill filling rate were optimized. To evaluate the influence of milling time on the yield of ellagitannin, the milling time was investigated at 5 min, 10 min and 15 min, respectively. With the extension of milling time, the peak areas of three compounds (corilagin, geraniin, and ellagic acid) generally showed a downward trend, shown in Figure 2A. The possible reason was that the longer milling time resulted in the accumulation of heat in the ball milling tank, which destroyed the compound structural units and led to the oxidation and partial decomposition of the ellagitannin compound. As a result, the milling time of 5 min was preferred.

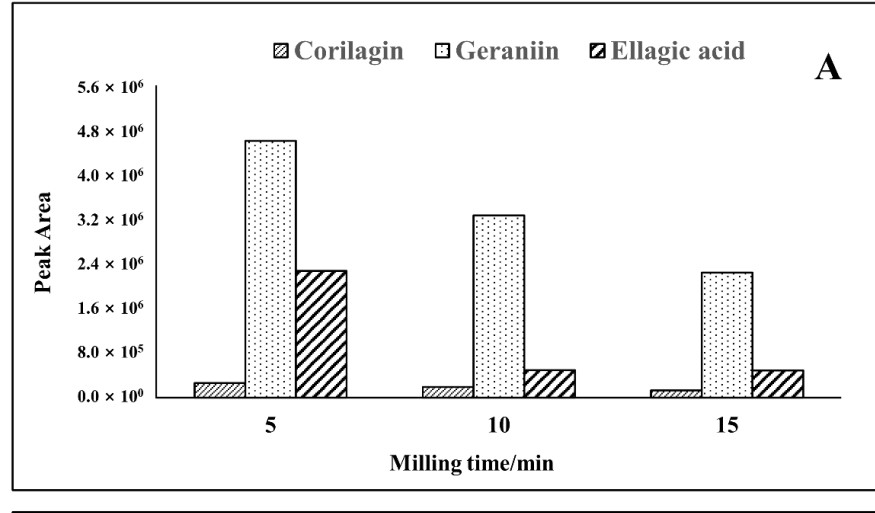

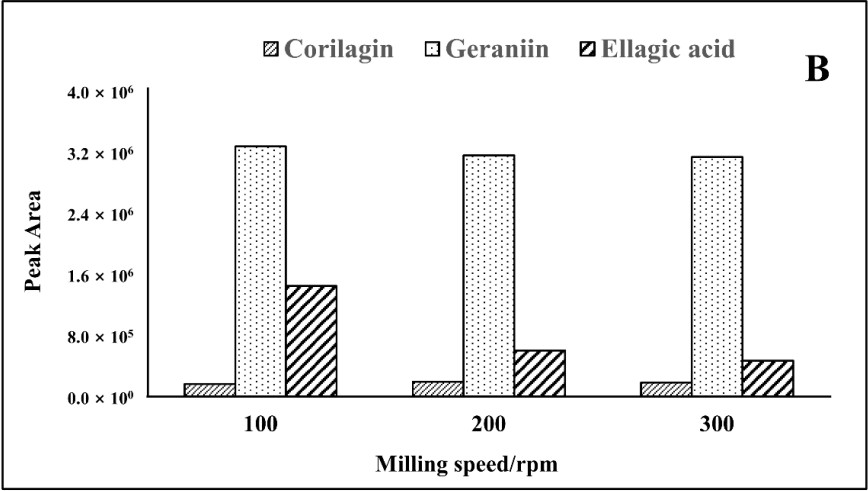

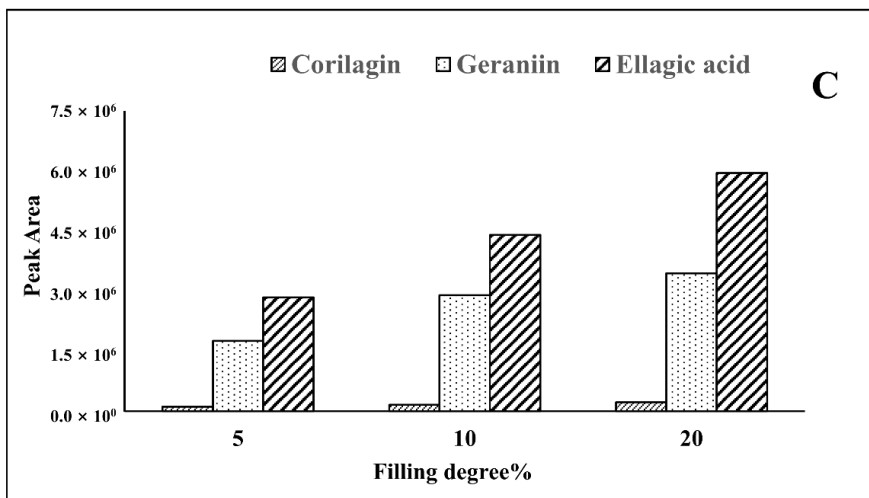

**Figure 2.** Effect of the MCAE procedure on the yield of ellagitannins and ellagic acid. (**A**) Effect of milling time; (**B**) Effect of milling speed; (**C**) Effect of filling degree.

Ball milling speed is a crucial factor for MCAE. The effect of ball milling speed on the extraction yield of ellagitannin from PUL was shown in Figure 2B. Three speeds including 100 rpm, 200 rpm and 300 rpm were investigated. It was clearly seen that at the ball milling speed of 100 rpm, the yield of the three compounds (corilagin, geraniin, and ellagic acid) reached the maximum value. With increases in ball milling speed, the peak areas

of corilagin and geraniin did not change significantly. Generally speaking, the MCAE of the ball mill can continuously destroy cell walls and promote the reaction of bioactive substances with solid phase reagents, thereby greatly improving the extraction efficiency. However, the ball milling speed did not further improve the entire extraction process at the speed of 200 rpm and 300 rpm. Hence, 100 rpm was chosen for the optimal ball milling parameter.

The ball mill filling rates were another factor that influenced the yield of ellagitannin. The filling rates at 5.2%, 10.5% and 20.9% were investigated and the results were shown in Figure 2C. Under the optimal milling time and ball milling speed, with the increase of the filling rate, the peak areas of geranium and ellagic acid showed an overall upward trend, while the peak area of ellagic acid increased significantly. It indicated that the mechanical force in the ball milling tank increased while the increased filling rate correspondingly increased the wall-breaking effect of the plant cells, and more ellagic acid was released. Moreover, the filling rate of 20.9% was used for the ball milling work.

### 3.2. Optimization of Acid Hydrolysis Conditions

Considering that ellagic acid is an acid hydrolysate of polymerized ellagitannin, the parameters of acid hydrolysis conditions, including acid hydrolysis reagent, acid hydrolysis temperature, and acid hydrolysis time were optimized in this work. Different acid hydrolysis reagents such as sulfuric acid, hydrochloric acid and formic acid were used. The yields of ellagic acid showed that sulfuric acid had a better acidolysis effect. Then, the concentration of sulfuric acid was further optimized. As shown in Figure 3A, the yield of ellagic acid increased with the increasing concentration of sulfuric acid, and the yield of ellagic acid was the highest when the concentration of sulfuric acid was 0.552 mol/L. When the concentration of sulfuric acid is higher than 0.552 mol/L, the yield of ellagic acid decreases. Hence, the concentration of sulfuric acid was selected as 0.552 mol/L.

The acid hydrolysis temperature exerted a greater impact on the yield of ellagic acid. Different acid hydrolysis temperatures at 20 °C, 30 °C, 40 °C, 50 °C, 60 °C, 70 °C, 80 °C and 90 °C were investigated as shown in Figure 3B. The highest peak area of ellagic acid was obtained when the acidolysis temperature was set at 40 °C. As the temperature increased, the yield of ellagic acid increased. This was because high temperatures may increase the degradation rate of ellagitannin. However, when the temperature rose to a certain level (40 °C), the yield of ellagic acid decreased. It was speculated that excessive temperature might cause the oxidative damage of ellagic acid.

Acid hydrolysis time showed little effect on the yield of ellagic acid, shown in Figure 3C. Ellagitannin could be degraded into ellagic acid within 30 min. When the acid hydrolysis time was longer than 30 min, it remained stable. 30 min was thought to be the better acid hydrolysis time.

A low concentration of sulfuric acid leads to inadequate acidolysis and a low yield of ellagic acid. However, the yield of ellagic acid decreased and the energy consumption increased with the high concentration of sulfuric acid. Considering the sensitivity of ellagic acid to air, the excessive temperature may accelerate the oxidation of ellagic acid. Finally, the acid hydrolysis conditions are determined as follows: the concentration of sulfuric acid was 0.552 mol/L, the acidolysis time was 30 min, and the acidolysis temperature was 40 °C.

The yield and content of crude ellagic acid could be improved significantly under the optimized acidolysis conditions. As the content of ellagic acid increased, the contents of corilagin and geraniin decreased. It may be speculated that ellagitannins were more likely to be broken down into ellagic acid in an acid environment [34,35]. In previous reports, Wei [36] and Li [37] studied the extraction process of ellagic acid from muscadine and red raspberry, respectively. The contents of ellagic acid were 616.21 µg/g and 322 µg/g, which were much lower than the 10.2 mg/g in this work.

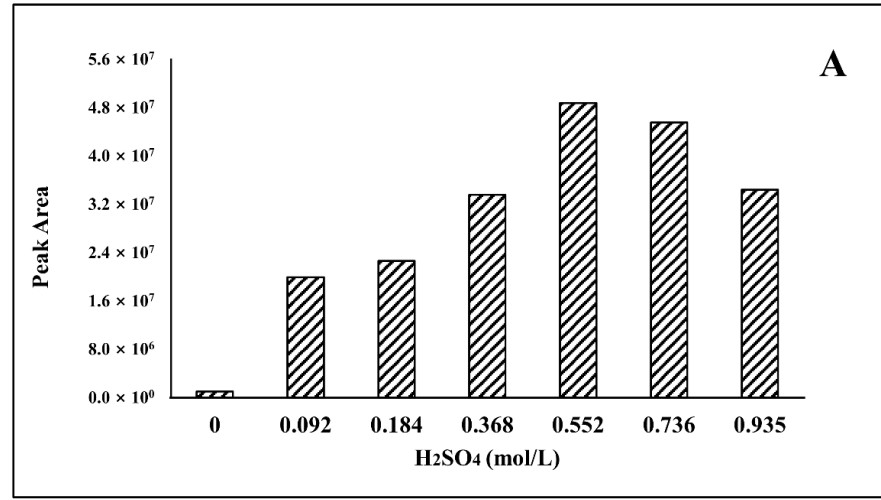

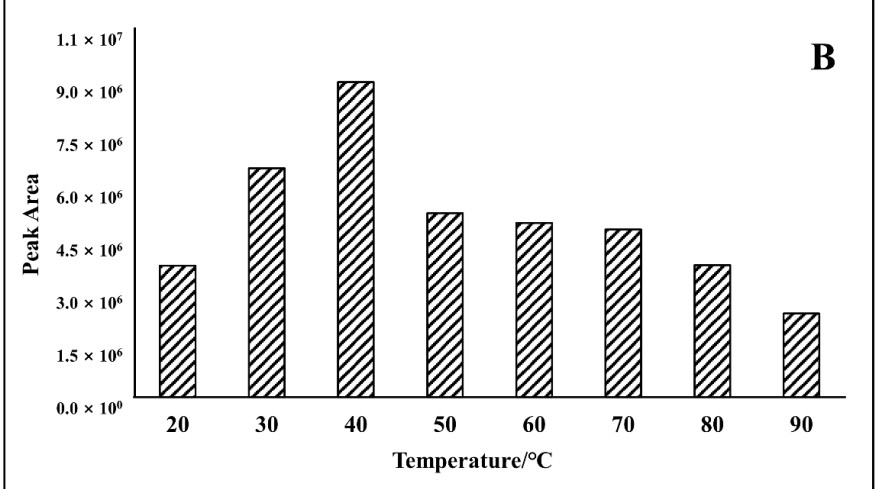

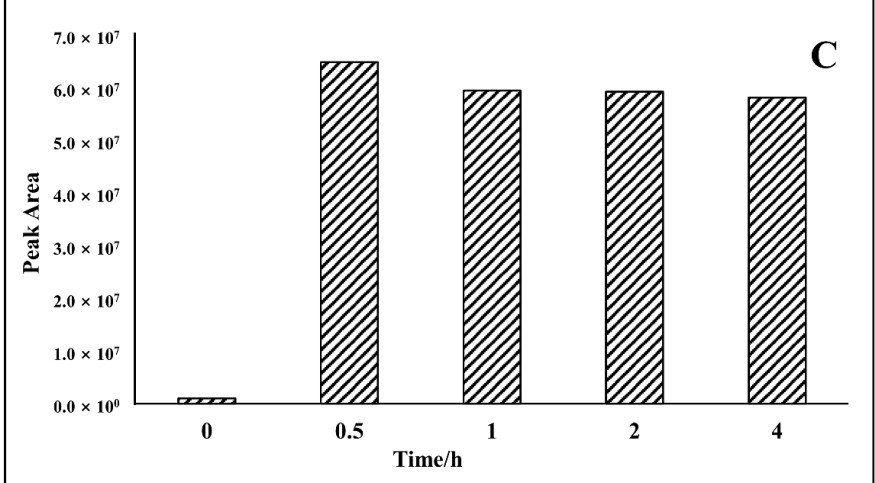

**Figure 3.** Effect of acid hydrolysis conditions. (**A**) Effect of $H_2SO_4$ concentration; (**B**) Effect of acid hydrolysis temperature; (**C**) Effect of acid hydrolysis time.

### 3.3. Screening of Optimum Resin

The adsorption and desorption of ellagic acid by different types of macroporous resins were shown in Table 2 and Figure 4. The XDA-8D type macroporous resin showed the highest adsorption rate (78.03%) of ellagic acid. Although its desorption rate was inferior to HPD450 and LXA-8 resins when the ethanol was selected as the adsorbent, XDA-8D

had a higher recovery rate than other resins. As a result, XDA-8D was selected for the separation and purification of crude ellagic acid.

**Table 2.** Results of the static adsorption and desorption of macroporous resins.

| Trade Name | Adsorbent Concentration (mg/mL) | Desorption Solution Concentration (mg/mL) | Adsorption Rate (%) | Desorption Rate (%) | Recovery Rate (%) |
|---|---|---|---|---|---|
| XAD-2 | 0.078 | 0.007 | 13.60 | 60.64 | 8.25 |
| HP-20 | 0.076 | 0.010 | 16.03 | 66.68 | 10.69 |
| AB-8 | 0.067 | 0.010 | 25.80 | 42.59 | 10.99 |
| XDA-8D | 0.021 | 0.016 | 78.03 | 74.47 | 58.11 |
| LSA-8D | 0.066 | 0.012 | 26.85 | 47.78 | 12.83 |
| HPD450 | 0.081 | 0.009 | 0.00 | 92.77 | 9.42 |
| HPD826 | 0.045 | 0.015 | 39.92 | 67.98 | 6.73 |
| DA201 | 0.062 | 0.017 | 31.00 | 62.44 | 19.36 |
| LXA-8 | 0.078 | 0.009 | 12.81 | 76.70 | 9.82 |
| LX-8 | 0.060 | 0.020 | 33.14 | 67.62 | 22.41 |

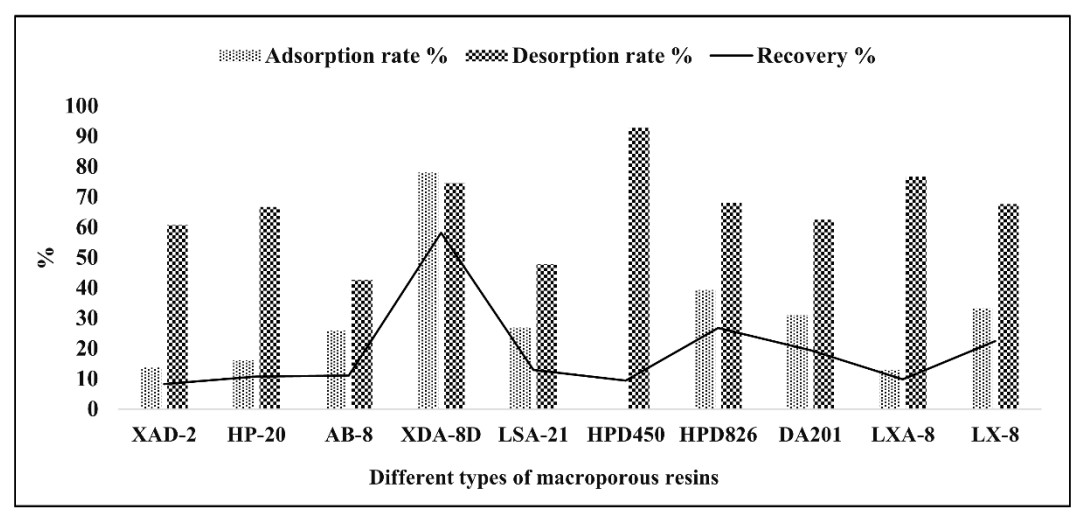

**Figure 4.** Effect of resin type on the static adsorption and desorption of ellagic acid.

### 3.4. Static Adsorption Kinetics and Adsorption Isotherms

To enrich ellagic acid from PUL extracts using microporous resins effectively, the optimum type of microporous resin was screened first. Because the external factors have a significant effect, the resin adsorption behavior is significantly influenced by many external factors. To evaluate these effects on the resin adsorption capacity, the adsorption kinetics tests and adsorption isotherms of XDA-8D microporous resin for crude ellagic acid were carried out (Figures 5 and 6).

According to Figure 5, the adsorption rate of XDA-8D for crude ellagic acid increased to the biggest within 1 h. Then, it still maintained a fast adsorption rate within 1–5 h. However, it proceeded slowly between 5–10 h and reached equilibrium after 10 h. The results showed that the adsorption capacity was about 1.2 mg/g of dry resin.

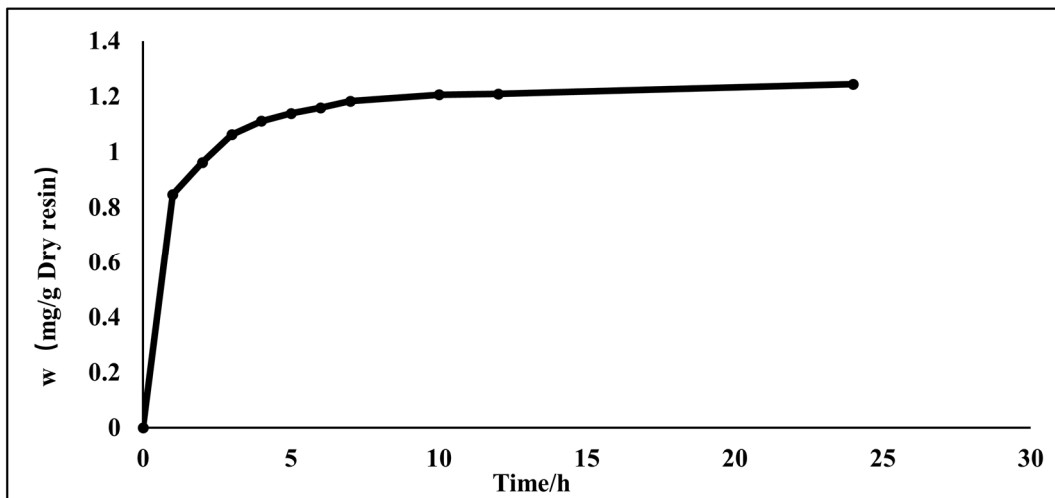

**Figure 5.** Static adsorption kinetics.

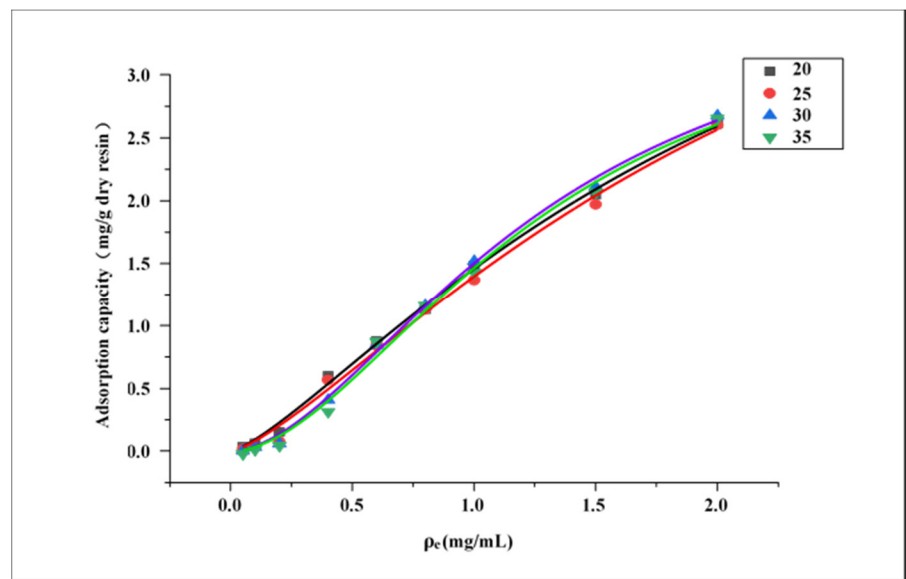

**Figure 6.** Static adsorption kinetics.

To study the effect of temperature on static adsorption, the adsorption isotherms were investigated at 20 °C, 25 °C, 30 °C and 35 °C. According to the typical adsorption isotherm classified by Brunauer [38] and the preferential equilibrium curves, the adsorption isotherm of XDA-8D belonged to the convex preferential adsorption isotherm (shown in Figure 6). This kind of adsorption isotherm is convex upward along the coordinate direction of adsorption capacity and is called preferential adsorption isotherm. According to Brunauer-Deming-Deming-Teller (BDDT) classification, this means that monolayer molecular adsorption occurs when the pore size of the adsorbent capillary is slightly larger than that of the adsorbent molecule. The experiment data of adsorption isotherms shown in Table 3 were well fitted to a Langmuir model [39]. It showed that the temperature had a certain influence on the adsorption process; 30 °C was the best, with an adsorption capacity of 2.462 mg/mL.

**Table 3.** Adsorption isotherm equation.

| T/°C | Langmuir Isotherm Equation | $R^2$ |
|------|---------------------------|-------|
| 20 | $Y = 1.994 \, X^{1.320}/(1 + 0.369 \, X^{1.320})$ | 0.9970 |
| 25 | $Y = 1.853 \, X^{1.346}/(1 + 0.329 \, X^{1.346})$ | 0.9932 |
| 30 | $Y = 2.462 \, X^{1.773}/(1 + 0.640 \, X^{1.773})$ | 0.9970 |
| 35 | $Y = 2.392 \, X^{1.825}/(1 + 0.636 \, X^{1.825})$ | 0.9927 |

### 3.5. Dynamic Adsorption and Elution

The crude ellagic acid product was adsorbed by XDA-8D macroporous resin and eluted with ethanol aqueous solution in gradients, collecting the 30% ethanol to 80% ethanol elution fraction. After removing the solvent, it was characterized by $^1$H NMR, $^{13}$C NMR and MS, shown in Figures S1–S3 (Supplementary Materials), respectively. Meanwhile, the content measured by UPLC was 97% (shown in Figure S4).

### 3.6. UPLC Quantitative Analysis

3.6.1. Linearity and Limits of Detection and Quantification

The calibration curves were plotted with a series of concentrations of standard solutions. Each analyte curve was made at six levels. Acceptable linear correlation and high sensitivity at these conditions were confirmed by the correlation coefficients ($R^2$, 0.9989–0.9999). The limits of detection (LODs) and limits of quantification (LOQs) for standards were estimated at signal-to-noise ratios (S/N) of three and ten, respectively, by injecting a series of dilute solutions with known concentrations. The detailed information regarding calibration curves, linear ranges, LODs and LOQs are displayed in Table S1.

3.6.2. Precision, Repeatability, Stability and Recovery

The precisions calculated as relative standard deviation (RSD) were within the range of 0.42–1.28%. The RSD values of three compounds were within the range from 4.67 to 6.15%, which revealed a high repeatability of the method. Stability of the sample solution was tested at room temperature in 12 h. The RSD values of three compounds were all within 5.37%, which demonstrated a good stability within the tested period.

The data for precision, repeatability, stability and recovery were also listed in Table S1. As shown in Table S1, the mean recovery rates of three compounds varied from 93.70 to 107.00% (RSD $\leq$ 7.10%).

### 4. Conclusions

In this study, a combined mechanochemical-macroporous resin adsorption method was established to separate and purify ellagic acid from PUL. The mechanochemistry ball milling coupled with the ultrasonic-assisted solvent extraction method was utilized to increase the extraction yield of ellagitannin. Under the optimized ball-milling conditions, the yield of ellagitannin was increased as compared to the traditional extraction method. The ellagitannin in the PUL extract could be converted into ellagic acid under sulfuric acid hydrolysis. The optimal reaction conditions were as follows: the sulfuric acid amount was 0.552 mol/L, the acidolysis time was 30 min, and the acidolysis temperature was 40 °C. Finally, the crude ellagic acid was separated and purified by XDA-8D macroporous resin to obtain ellagic acid. Under the optimal technological conditions, the yield of ellagic acid was 10.2 mg/g, and the content was over 97%. It is rapid and efficient for the preparation of ellagic acid. Meanwhile, it can also provide a technical basis for the comprehensive utilization of PUL.

**Supplementary Materials:** The following are available online at https://www.mdpi.com/article/10.3390/separations8100186/s1, Figure S1: The $^1$H NMR spectrum of ellagic acid. Figure S2: The $^{13}$C NMR spectrum of ellagic acid. Figure S3: The MS spectrum of ellagic acid. Figure S4: UPLC chromatogram of ellagic acid. Table S1: Calibration curves, linear ranges, LODs, LOQs, precision, repeatability, stability and recovery of 5 standards compounds.

**Author Contributions:** Conceptualization, X.L.; methodology, S.X.; software, Z.G. and S.X.; validation, Z.G.; formal analysis, Z.G. and S.X.; investigation, Y.X.; resources, S.X.; data curation, S.X.; writing—original draft preparation, Z.G.; writing—review and editing, Z.G., Y.X. and X.L.; project administration, Z.G. and X.L. All authors have read and agreed to the published version of the manuscript.

**Funding:** This research received no external funding.

**Institutional Review Board Statement:** Not applicable.

**Informed Consent Statement:** Not applicable.

**Acknowledgments:** This work was supported by the cooperative project with Zhejiang Hisoar Pharmaceutical Co., Ltd. (KYY-HX-20180525) and the scientific research project of the Zhejiang Provincial Department of Education (Y202043200).

**Conflicts of Interest:** The authors declare no conflict of interest.

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
