# Peer review of "The Separation and Purification of Ellagic Acid from Phyllanthus urinaria L. by a Combined Mechanochemical-Macroporous Resin Adsorption Method"

_separations, doi:10.3390/separations8100186_

Round 1

Reviewer 1 Report

The authors have selected an interesting subject and made an effort to study it. However, the MS lacked important information, especially in the M&M section and in presentation of results.

More detailed comments below.

Abstract
22 low cost-consuming ->low-cost Rephrase the sentence for clarity.

Keywords: Consider adding another suitable keyword

1. Intro
27 remove "structure is given below"
27 the -> an
35 AD-mediated: explain abbreviation when first mentioned
62 was -> has been

72 Although UPLC is a common abbreviation, you should explain it when first mentioned
92 meshes -> mesh
131 every 1 hour -> every hour
144 80%, and
151 which in hence increase -> which increases
186-189 Can you rephrase these?
The graphs should be better in quality
200 Peak are -> peak area
201 Give a more thorough explation for the figure
218 screened firstly -> screened first.

Fig.6 Explain the figure more thoroughly.

The methods should be described more thoroughly with methods and method validation data.
Where are the NMR results?

Author Response

Dear reviewer,

We would like to thank you for the careful reading of our manuscript (separations-1395866) entitled “Separation and purification of ellagic acid from Phyllanthus urinaria L. by a combined mechanochemical-macroporous resin adsorption method”. According to the comments, we have carefully revised the manuscript and found these comments are helpful in improving the quality of the manuscript. Listed below are the reviewers' comments and the changes that were made in the manuscript.

Reviewer 2 Report

The paper titled "Separation and purification of ellagic acid from Phyllanthus urinaria L. by a combined mechanochemical-macroporous resin adsorption method” is very interesting and well written.

The authors have developed a method to extract (mechanochemical-assisted extraction and macroporous adsorption resin), separate and purify ellagic acid from Phyllanthus urinaria L.

Personally, I recommend the publication in this journal only after a few revisions.

 Specifically, I advise reformulating the "material and methods" paragraph, it is a bit confusing. The authors are asked to describe the extraction procedures (before) and the analytical methodologies (after).

Moreover, I advise you minor spell-check of the English language.

Author Response

(The authors gave the same response as above.)

Reviewer 3 Report

The authors could provide the type of detection after UPLC.

A discussion comparing the obtained results with previous published data could be presented.

Author Response

(The authors gave the same response as above.)

Round 2

Reviewer 1 Report

The authors have responded to reviewers´ comments and questions and made changes based on them. However, I´d still liked to add more comprehenseive explanations to figures to fully describe their contents.

Additionally, a small comment: I still think the mesh size should be presented as "80 mesh".

Author Response

Dear reviewer,

We really appreciate your consideration for giving us the opportunity to resubmit our manuscript (separations-1395866) entitled “Separation and purification of ellagic acid from Phyllanthus urinaria L. by a combined mechanochemical-macroporous resin adsorption method”. According to the comments, we have carefully revised the manuscript and found these comments are helpful in improving the quality of the manuscript. Listed below are the reviewers' comments and the changes that were made in the manuscript.

Question 1: The authors have responded to reviewers´ comments and questions and made changes based on them. However, I´d still liked to add more comprehenseive explanations to figures to fully describe their contents.

Answer: Thank you for your suggestion. We have made the revisions to describe the pictures as comprehensive as possible. The modifications were as follows:

  • “and leaded to the oxidation and partial decomposition of ellagitannin compound.” was added in line 190-191 to explain the Figure 2(A).
  • “Ball milling speed is a crucial factor for MCAE.” was added in line 192 to explain the Figure 2(B).
  • “It was clearly seen that, at the ball milling speed of 100 rpm, the yield of the three compounds (corilagin, geraniin, ellagic acid) reached the maximum value. With increases in ball milling speed, the peak areas of corilagin and geraniin did not change significantly. Generally speaking, MCAE of the ball mill can continuously destroy cell wall and promote the reaction of bioactive substances with solid phase reagents, thereby greatly improving the extraction efficiency. However, the ball milling speed did not further improve the entire extraction process at the speed of 200 rpm and 300 rpm.” was changed to “With increases in ball milling speed, the peak areas of corilagin and geraniin did not change significantly, while the peak area of ellagic acid decreased significantly due to the high energy generated by the high speed leading to the degradation of the compound.” in line 194-203 to explain the Figure 2(B).
  • “As the temperature increased, the yield decreased. It was speculated that high temperatures might cause the degradation of ellagic acid.” was changed to “As the temperature increased, the yield of ellagic acid increased. This was because high temperature may increase the degradation rate of ellagitannin. However, when the temperature rose to a certain level (40°C), the yield of ellagic acid decreased. It was speculated that excessive temperature might cause the oxidative damage of ellagic acid.” in line 234-239 to explain the Figure 3(B).

Question 2: Additionally, a comment: I still think the mesh size should be presented as "80 mesh".

Answer: “80 meshes” was changed to “80 mesh” in line 22.
